# The Rationale for the Dual-Targeting Therapy for RSK2 and AKT in Multiple Myeloma

**DOI:** 10.3390/ijms23062919

**Published:** 2022-03-08

**Authors:** Reiko Isa, Mano Horinaka, Taku Tsukamoto, Kentaro Mizuhara, Yuto Fujibayashi, Yoko Taminishi-Katsuragawa, Haruya Okamoto, Shusuke Yasuda, Yuka Kawaji-Kanayama, Yayoi Matsumura-Kimoto, Shinsuke Mizutani, Yuji Shimura, Masafumi Taniwaki, Toshiyuki Sakai, Junya Kuroda

**Affiliations:** 1Division of Hematology and Oncology, Department of Medicine, Kyoto Prefectural University of Medicine, Kyoto 602-8566, Japan; isa-r@koto.kpu-m.ac.jp (R.I.); ttsuka@koto.kpu-m.ac.jp (T.T.); ken-mizu@koto.kpu-m.ac.jp (K.M.); f-yuto@koto.kpu-m.ac.jp (Y.F.); yoko-k@koto.kpu-m.ac.jp (Y.T.-K.); hokamoto@koto.kpu-m.ac.jp (H.O.); yuka0914@koto.kpu-m.ac.jp (Y.K.-K.); m-yayoi@koto.kpu-m.ac.jp (Y.M.-K.); mizushin@koto.kpu-m.ac.jp (S.M.); yshimura@koto.kpu-m.ac.jp (Y.S.); taniwaki@koto.kpu-m.ac.jp (M.T.); 2Department of Drug Discovery Medicine, Graduate School of Medical Science, Kyoto Prefectural University of Medicine, Kyoto 602-8566, Japan; m-hori@koto.kpu-m.ac.jp (M.H.); syu-ysd@koto.kpu-m.ac.jp (S.Y.); tsakai@koto.kpu-m.ac.jp (T.S.); 3Center for Molecular Diagnostics and Therapeutics, Kyoto Prefectural University of Medicine, Kyoto 602-8566, Japan

**Keywords:** AKT, gene set enrichment analysis, mTOR, multiple myeloma, MYC, RSK2

## Abstract

Multiple myeloma (MM) is characterized by remarkable cytogenetic/molecular heterogeneity among patients and intraclonal diversity even in a single patient. We previously demonstrated that PDPK1, the master kinase of series of AGC kinases, is universally active in MM, and plays pivotal roles in cell proliferation and cell survival of myeloma cells regardless of the profiles of cytogenetic and genetic abnormalities. This study investigated the therapeutic efficacy and mechanism of action of dual blockade of two major PDPK1 substrates, RSK2 and AKT, in MM. The combinatory treatment of BI-D1870, an inhibitor for N-terminal kinase domain (NTKD) of RSK2, and ipatasertib, an inhibitor for AKT, showed the additive to synergistic anti-tumor effect on human MM-derived cell lines (HMCLs) with active RSK2-NTKD and AKT, by enhancing apoptotic induction with BIM and BID activation. Moreover, the dual blockade of RSK2 and AKT exerted robust molecular effects on critical gene sets associated with myeloma pathophysiologies, such as those with MYC, mTOR, STK33, ribosomal biogenesis, or cell-extrinsic stimuli of soluble factors, in HMCLs. These results provide the biological and molecular rationales for the dual-targeting strategy for RSK2 and AKT, which may overcome the therapeutic difficulty due to cytogenetic/molecular heterogeneity in MM.

## 1. Introduction

Multiple myeloma (MM) is a clonal plasma cell neoplasm, and is the second most common hematologic malignancy [1]. Individual patients’ MM disease features are built up by several cooperative components. Cytogenetic/genetic/epigenetic instability leads to the stepwise and branching clonal evolution in myeloma cells, resulting in multiple molecularly heterogeneous subclones [2,3,4,5,6,7,8,9]. The activation of oncogenic driver signals, such as MYC, RAS/ERK, PI3K/AKT, and NF-κB, as well as the dysfunction/loss of pivotal tumor suppressors, such as TP53 or CDKN2C, cooperatively and sometimes compensatively promote the disease progression, the aggressiveness, and treatment-resistant phenotype of tumor cells [3,6,10,11,12,13,14,15]. Cell-extrinsic tumor microenvironment supports the survival and proliferation of myeloma cells via adhesion molecules, soluble factors, or extracellular vesicles, and also protects myeloma cells from cytotoxic insults [6,16,17,18], whereas the disruption of tumor immune surveillance system allows disease expansion [19,20,21,22,23,24].

To combat MM, which has remarkable cytogenetic/molecular heterogeneity among patients and intraclonal diversity even in a single patient, a series of therapeutic strategies has been developed over the past two decades. Thanks to major players including proteasome inhibitors (PIs), immune therapeutics immunomodulatory drugs (IMiDs), and monoclonal antibodies (MoAbs) against CD38 or SLAMF-7, the treatment outcome of MM has been greatly improved [25,26]; however, the disease is still mostly incurable even with the latest cellular immunotherapy, such as chimeric antigen T-cell receptor therapy and bispecific T-cell engager [27,28]. As a result, there is still an unmet medical need for the development of a new class of agents that are universally active against myeloma cells regardless of cytogenetic/molecular heterogeneity.

We previously demonstrated that phosphoinositide-dependent protein kinase-1 (PDPK1) is constitutively and universally active in myeloma cells from the majority of patients, and plays pivotal roles in cell proliferation and cell survival of myeloma cells regardless of the profiles of cytogenetic and genetic abnormalities [29]. In myeloma cells, PDPK1 is overexpressed mainly by the epigenetic repression of miR-375 [30], maintaining its constitutively active status through autophosphorylation. However, giving PDPK1 broad function as the master regulator of at least 23 AGC kinases, such as RSKs, p70S6K, AKT, PKC, and ILK, as well as the regulator of various non-AGC kinases, such as PLK-1, β3-integrin, or ROCK1, therapeutic inhibition of PDPK1 could result in a variety of unwanted off-target effects [31]. As a result, for the development of an effective and less toxic therapeutic strategy, the optimal selection of molecular targets from a series of PDPK1 substrates is required. Among the various PDPK1 substrates, we believe ribosomal S6 kinase 2 (RSK2) is one of the best candidates [32]. In physiologic conditions, PDPK1 activates the N-terminal kinase domain (NTKD) of RSK2 through phosphorylation so that RSK2 plays the essential role as the signaling hub of the RAS/RAF/ERK pathway. Importantly, while the activation of PDPK1 is visible in clonal plasma cells of asymptomatic MM, the activation of RSK2-NTKD becomes more visible when the disease progresses to the symptomatic phase, suggesting the involvement of RSK2 activation in MM progression to the treatment-necessary disease stage [29]. Furthermore, the inactivation of RSK2-NTKD causes apoptosis in myeloma cells regardless of their cytogenetic/molecular features [32], while there is a compensative interaction between RAS/ERK/RSK2-mediated signal and AKT, which is another major oncogenic substrate of PDPK1 in various types of cancer cells [33,34,35,36]. These findings prompted us to investigate the therapeutic efficacy and mechanism of action of dual blockade of two major PDPK1 substrates, RSK2-NTKD and AKT, in MM.

## 2. Results

### 2.1. Effects of BI-D1870 and Ipatasertib in 6 Human Myeloma Cell Lines (HMCLs)

We first looked at the baseline activation status of PDPK1, RSK2, AKT, and ERK1/2 in six HMCLs. PDPK1 and RSK2 were found to be activated through phosphorylation in all six HMCLs examined, while the activation statuses of ERK and AKT were variable among HMCLs. AKT was found to be active in NCI-H929, OPM-2, KMS-12-BM, and KMS-28-PE cells, while only faint expression of p-AKT^Ser473^ was detected in AMO-1 and RPMI8226 cells (Figure 1A). Following that, we examined the growth inhibitory effects of BI-D1870, a specific inhibitor for RSK2-NTKD, and ipatasertib, an inhibitor for AKT, in six HMCLs. The treatment with BI-D1870 showed the dose-dependent growth inhibitory effect on all HMCLs examined (Figure 1B). In contrast, we observed cellular context-dependent growth inhibition by ipatasertib in HMCLs studied; while ipatasertib exerted the dose-dependent growth inhibitory effect on four HMCLs of NCI-H929, OPM-2, KMS-12-BM, and KMS-28-PE cells with active p-AKT^Ser473^, it has no or only modest growth inhibitory effect on AMO-1 and RPMI8226 cells without active AKT (Figure 1A,C). In addition, with these growth inhibitory effects, treatment with BI-D1870 repressed the phosphorylation of RSK2^Ser227^, while treatment with ipatasertib for 48 h exerted the optimal inhibition of AKT activity as shown by the dephosphorylation of PRAS40^Thr246^, which is the substrate of AKT. Unlike other cancers [33,34,35,36], both BI-D1870 and ipatasertib showed no cross-reaction to their respective target molecules. In addition, the combination of the two agents did not result in further inactivation of RSK2 and AKT compared with the effect induced by a single agent in the two HMCLs (Figure 1D).

### 2.2. The Combinatory Anti-Tumor Effect of BI-D1870 and Ipatasertib on HMCLs with Active RSK2-NTKD and AKT

The combination of BI-D1870 and ipatasertib was then tested in four RSK2-NTKD- and AKT-activated HMCLs. The normalized isobologram analysis revealed that the two agents had additive to synergistic growth inhibitory effects (Figure 2A), and also showed favorable dose-reduction indices (DRIs) of BI-D1870 and ipatasertib at most effect levels in their combination setting (Appendix A). In addition, the combination of BI-D1870 and ipatasertib showed significantly more powerful growth suppression compared to the effect by either ipatasertib or BI-D1870 alone, not only in HMCLs, but also in primary myeloma cells from a patient who acquired resistance to PI, IMiDs, and anti-CD38 MoAb (Figure 2B). Cell cycle analysis revealed that BI-1870 treatment increased the proportion of cells in the sub G1 fraction, implying the induction of apoptosis, whereas ipatasertib treatment blocked the G1/S transition without increasing cells in the sub G1 fraction. However, when ipatasertib was added to BI-D1870, there was significantly increased cells in sub G1 fraction compared to that induced by BI-D1870 alone (Figure 2C). This was also confirmed by the significant increase in Annexin-V-positive cells, when ipatasertib was added to BI-D1870 compared to that induced by BI-D1870 alone (Figure 2D,E). These findings suggest that dual targeting of RSK2-NTKD and AKT has a favorable combinatory anti-tumor effect in HMCLs when it is accompanied by apoptosis induction.

### 2.3. Mechanisms of Apoptosis Induced by the Combination of BI-D1870 and Ipatasertib in HMCLs

We next investigated the mechanisms underlying apoptosis induction by the combination of BI-D1870 and ipatasertib in HMCLs. Results of Caspase assays showed that, while ipatasertib did not cause the activation of caspases, the addition of ipatasertib onto BI-D1870 resulted in significant enhancement of the activities of Caspase-3/7, Caspase-8, and Caspase-9 compared to those induced by BI-D1870 alone (Figure 3A). Western blot (WB) analyses also revealed the more prominent processing of Caspase-3, Caspase-8, and Caspase-9, and the resultant increase in cleaved PARP by the combination of the two agents compared to that induced by BI-D1870 alone (Figure 3B and Appendix A). These also implicated the involvement of both cell-intrinsic and extrinsic apoptotic pathways in the anti-tumor effect by the dual targeting of RSK2-NTKD and AKT.

With these results, we then investigated the expression changes in apoptosis regulators. As a result, the combination of BI-D1870 and ipatasertib resulted in the changes in two major pro-apoptotic BH3-only proteins, BIM and BID, as shown by the increased expression of three BIM isoforms and the emergence of truncated BID (t-BID) compared with those induced by either BI-D1870 or ipatasertib alone in two HMCLs examined (Figure 3C and Appendix A). BIM induction was at least partially mediated by transcriptional activation (Figure 3D). However, in NCI-H929 cells, knocking down either BIM or BID did not completely prevent the induction of apoptosis by the combination of BI-D1870 and ipatasertib (Appendix A). In addition, we found no prominent/meaningful expression change in other BH3-only proteins, such as BAD or PUMA, BAX, anti-apoptotic proteins, such as BCL-2, BCL-X_L_, and MCL-1, and series of molecules involved in death receptor-mediated apoptosis pathways, such as DR4, DR5, FAS, TNFR1, FLIP, and TRAIL (Appendix A). These findings suggested the involvement of an unknown mechanism in apoptosis induced by the concomitant blockade of RSK2-NTKD and AKT.

### 2.4. Molecular Effects Induced by the Dual Blockade of RSK2-NTKD and AKT

To understand the molecular mechanism underlying the anti-MM effect by the dual blockade of RSK2-NTKD and AKT, the study performed gene expression profile (GEP) analysis to comprehensively analyze commonly modulated genes by the combinatory treatment with BI-D1870 and ipatasertib in comparison with those modulated either by BI-D8170 or ipatasertib alone in NCI-H929 and OPM-2 cells (Appendix A). Overall, the correlation coefficients for gene expression changes between NCI-H929 and OPM-2 cells were 0.23, 0.16, and 0.30 after the treatment by BI-D1870, ipatasertib, and the combination of two agents, respectively, indicating the existence of weak correlation only with the combination of treatment, but not with single agents between the two HMCLs (Figure 4A). However, as shown in Figure 4B, 144, and 78, genes were found to be commonly upregulated by BI-D1870 and ipatasertib respectively, while 62 and 72 genes were commonly downregulated by BI-D1870, and ipatasertib, in the two HMCLs. There were only 38 upregulated and 19 downregulated genes that overlap between genes modulated by the sole treatment of BI-D1870 or ipatasertib. It also noted that, 176 genes were commonly upregulated, while 194 genes were commonly downregulated by the combination of BI-D1870 and ipatasertib in the two HMCLs. Importantly, 94 upregulated genes and 135 downregulated genes were significantly modulated only when HMCLs were exposed to combinatory treatment, indicating the occurrence of more molecular changes only when HMCLs were exposed to the combinatory treatment of BI-D1870 and ipatasertib (Figure 4B). While the overall molecular response to the inhibition of RSK2-NTKD, AKT, and dual blockade of both kinases is variable and cellular context-dependent at the transcriptional level, the findings suggest the induction of more common molecular impact by the dual blockade of RSK2 and AKT compared to that induced by the blockade of either RSK2 or AKT alone. We also validated the GEP results by adding quantitative reverse transcription-polymerase chain reaction (qRT-PCR) for the expression of several randomly selected genes (Appendix A).

Then, we used gene set enrichment analysis (GSEA) to compare gene expression changes caused by RSK2-NTKD and AKT blockade to changes caused by RSK2 or AKT blockade alone. In brief, we first performed gene clustering in NCI-H929 and OPM-2 cells using BI-D1870, ipatasertib, and their combination treatments (Figure 4C and Table 1).

As judged by the false discovery rate q-value, various gene sets were found to be more significantly modulated by the combination of BI-D1870 and ipatasertib, compared with the treatment by either BI-D1870 or ipatasertib alone. Significant gene sets downregulated by the combination of two agents (Cluster C1 in Figure 4C; Table 1) included those for various oncogenic signatures, such as gene sets driven by upregulation of MYC and mTOR, and re-exposure to serum, interleukin (IL)-2 or IL-15 after starvation, and gene sets inactivated by gene knockdown of RPS14, the blockade of vascular endothelial growth factor (VEGF), and cellular differentiation, among others (Figure 5A,B; Table 2). Significant gene sets upregulated by the combination of two agents (Cluster C5 in Figure 4C; Table 1) included those downregulated by mTOR and MYC and upregulated by the knockdown of STK33, a pro-tumor serine–threonine kinase (Figure 5C,D; Table 2). Indeed, the co-treatment of BI-D1870 and ipatasertib caused the moderate inhibition of phospho-mTOR and the reduction in c-MYC expression (Figure 6).

## 3. Discussion

RAS/RAF/ERK and PI3K/AKT pathways are frequently activated by both various cell-intrinsic oncogenic events and extrinsic stimuli, and they play critical roles in myeloma pathophysiology. However, the clinical development of pathway-directed therapeutics targeting either RAS, RAF, PI3K, or AKT as the single target has failed in yielding results until today [37,38]. We are of the opinion that, although RSK2 is the bottleneck output hub of RAS/RAF/ERK pathway [39,40], the therapeutic effect of single targeting of RAS or RAF might be hampered by the PDPK1-mediated constitutive active status of RSK2, which is independent of RAS/RAF/ERK activity in myeloma cells, while the blockade of AKT alone has only cytostatic effects but no major cell killing effect, as was found in this study in myeloma cells.

In this study, we discovered that the combination of BI-D1870 and ipatasertib had a synergistic antiproliferative effect in HMCLs with active RSK2-NTKD and AKT. Although ipatasertib alone is not effective in inducing apoptosis in HMCLs, the addition of ipatasertib onto BI-D1870 significantly augmented the cell death-inducing effect with apoptosis, which was accompanied by the enhanced activation of both cell-intrinsic and -extrinsic apoptosis pathways. Indeed, dual blockade of RSK2 and AKT enhanced the transcriptional upregulation of BIM in HMCLs examined, those were consistent with other cancerous diseases [41]. The molecular effect by the combination of BI-D1870 and ipatasertib presumably promoted Casease-9 activation and led to the resultant activation of the cell-intrinsic apoptosis pathway. In addition, BI-D1870 treatment caused the processing of Caspase-8 to its cleaved form in HMCLs, which was consistent with a previous report showing that the blockade of RSK2 abrogates Caspase-8 stability [42]. However, it is unclear how the addition of ipatasertib onto BI-D1870 increased Caspase-8 activity remains to be verified. Indeed, Caspase-8 activation was not observed by ipatasertib treatment alone. We hypothesize that Caspase-3 may be involved in Caspase-8 feedback activity to promote proteolytic processing [43].

At the molecular level, no significant correlation in gene expression changes after RSK2 or AKT blockade was observed between the two HMCLs studied, which may reflect molecular heterogeneity between the two HMCLs studied. Despite this, a positive correlation of gene expression changes emerged following the combinatory blockade of the two molecules in the same two HMCLs. Moreover, the combinatory blockade of RSK2 and AKT caused gene expression changes in a larger number of genes, compared to those induced by the blockade of either RSK2 or AKT alone. Thus, the combining blockade of RSK2 and AKT has more robust molecular effects on HMCLs, raising the expectation for the universal efficacy on myeloma cells of diverse molecular features from different patients.

With this setting, GSEA disclosed that, in comparison with the sole blockade of either RSK2 or AKT, the combination of blockade in the two kinases proved more powerful effects on several critical gene sets involved in myeloma pathophysiologies, such as those associated with MYC, mTOR, STK33, ribosomal biogenesis, or cell-extrinsic stimuli of soluble factors. MYC is a multifunctional oncogenic transcription factor that affects various cell biological processes, including cell proliferation, apoptosis, and DNA damage in various cancerous diseases, including myeloma [44,45]. Furthermore, MYC is frequently overexpressed through various overlapping mechanisms, such as cytogenetic, epigenetic, and IRF4-mediated transcriptional mechanisms, in myeloma cells, and plays critical roles in cell survival and proliferation of myeloma cells, thereby constituting the fundamental process in both the development and progression of MM [45,46,47,48]. The therapeutic efficacy of IMiDs, such as lenalidomide and pomalidomide, has been known to be partly mediated by the repression of MYC through the blockade of IKZF family proteins and ARID2 [49,50]. Given a serious prognostic consequence of acquiring of the resistance to IMiDs-containing therapy in MM treatment, it is expected that the dual blockade of RSK2 and AKT constitutes a new strategy that potently overcomes IMiDs resistance in MM. mTOR activation by cell-extrinsic stimuli, such as by IL-6, VEGF, and insulin-like growth factor-1, also contribute to the disease progression of MM by promoting cell survival, cell metabolism, the cross-talk with the ubiquitin–proteasome system, and angiogenesis [51,52,53,54]. With this regard, it was reasonable to note that the dual blockade of RSK2 and AKT can also cause the downregulation of gene sets associated with cell-extrinsic stimuli (serum, IL-2, IL-15, and VEGF). Given that the inhibition of mTORC2 has the synergistic anti-MM effects with a proteasome inhibitor and IMiD [51,52], the dual blockade of RSK2 and AKT may be expected to enhance the effects of proteasome inhibitors and IMiDs on myeloma cells. It was also appealing that the simultaneous inhibition of RSK2 and AKT causes the downregulation of genes associated with RPS14, which is an essential component of ribosomal biogenesis. Although the pathologic role of abnormal expression of RSP14 has not been known in MM, the blockade of RSP14 has been shown to exhibit a tumor-suppressive role regardless of TP53 status in solid and hematologic cancers [55]. STK33 has been known to participate in the transcriptional activation of c-MYC [56], and also contributes to the angiogenetic process as the mediator of hypoxia-inducible factor-1α/VEGF signaling [57]. Because the functional involvement of STK33 in MM has been unknown, more research is needed to clarify its role.

Despite the remarkable improvement of prognosis of patients with MM by the advent of combinatory therapies of PIs, IMiDs, and monoclonal antibodies, MM still remains to be mostly incurable. Therefore, the therapeutic strategy that can overcome the resistance to the combination of the three class agents is an unmet medical need in MM. Reportedly, the incidence of mutations within the RAS/ERK pathways is significantly increased in MM refractory to PIs and IMiDs [58]. Mechanistically, oncogenic activation of RAS/RAF/ERK signaling is one of the mechanisms that enhance proteasome capacity, while treatment with PI causes the phosphorylation of AKT [59,60], and both mechanisms contribute to the resistance to proteasome inhibition in myeloma cells. AKT-mediated sustained eIF4E expression and C/EBP translation have been proposed as the resistant mechanisms for IMiDs [61]. Moreover, the resistant mechanisms for current standard therapeutics for MM include aberrant MYC overexpression [50], and the increased rate of protein translation [62]. Considering these, one of the clinical positions of the dual-targeting therapy for RSK2 and AKT may be theoretically the salvage setting after treatment failure of PIs, and/or IMiDs with or without monoclonal antibodies for CD38 or SLAMF-7. Unfortunately, we were not able to examine the in vivo anti-myeloma effect of the combination of BI-D1870 and ipatasertib, due to the uncertain in vivo performance of BI-D1870, and also due to the lack of a promising in vivo bioavailable inhibitor specific for RSK2-NTKD. This was the limitation of the current study, which urges the generation of in vivo bioavailable inhibitor for RSK2-NTKD. To implement this strategy in a clinical environment, the clinical development of bioavailable dual inhibitors for RSK2 and AKT is expected for MM [63].

In conclusion, our study revealed that the dual blockade of RSK2 and AKT exhibits anti-tumor effects in myeloma cells via the regulation of multidirectional cell-intrinsic and extrinsic molecular mechanisms for various critical molecules and biologic processes in myeloma development and progression, and, thus, is an attractive candidate of new therapeutic strategy for MM with diverse molecular backgrounds.

## 4. Materials and Methods

### 4.1. Cells and Reagents

This study used 6 HMCLs, NCI-H929, OPM-2, AMO-1 (Deutsche Sammlung von Mikroorganismen und Zellkulturen GmbH, Braunschweig, Neddersassen, Germany), RPMI8226 (American Type Culture Collection, Manassas, VA, USA), KMS-12-BM and KMS-28-PE (kind gifts from Dr. Ohtsuki T (Kawasaki Medical School, Kurashiki, Okayama, Japan)), harboring various types of chromosomal abnormalities and gene mutations (Appendix A). HMCLs were grown in RPMI-1640 with 10% fetal calf serum, 2 mM L-glutamate, and penicillin/streptomycin at 37 °C in a fully humidified atmosphere of 5% CO_2_ in the air. Patient-derived CD138-positive myeloma cells were isolated using MACSprep^TM^ Multiple Myeloma CD138 MicroBeads (Miltenyi Biotec, Bergisch Gladbach, Nordrhein-Westfalen, Germany) [5,29], and were grown in RPMI-1640 with 10% fetal calf serum, 2 mM L-glutamate, penicillin/streptomycin, and CellXVivo Human B Cell Expansion Kit (R&D systems, Minneapolis, MN, USA). The agents used were BI-D1870, which selectively inhibits the phosphorylation of RSK2^Ser227^ at NTKD in an ATP-competitive manner [64] (Cayman Chemical Company, Ann Arbor, MI, USA), and ipatasertib, which binds to AKT in an ATP-competitive manner and prevents substrate phosphorylation [65] (Selleck Biotech Ltd., Tokyo, Japan).

### 4.2. Cell Proliferation Assay

Cells were seeded in a flat-bottomed 96-well plate at 2.0 × 10^5^ cells/mL in 200 μL medium per well. After 48 h of treatment with various concentrations of BI-D1870, ipatasertib, or their combinations for the absolute number of viable cells was analyzed with Cell Counting Kit-8 (CCK-8) assay (Dojindo Molecular Technologies, Kamimashiki, Kumamoto, Japan) using a SpectraMax iD3 multi-mode microplate reader (Molecular Devices, San Jose, CA, USA). Combination index values and DRIs were analyzed using CalcuSyn software (Biosoft, Cambridge, Cambridgeshire, UK). Synergism is defined as more than the expected additive effect with CI < 1. DRI = 1 indicates no dose reduction, whereas DRI > 1 and <1 indicate favorable and unfavorable dose-reduction, respectively [66].

### 4.3. Analysis of Cell Cycle and Detection of Apoptosis

After 48 h of treatment with an agent(s), cells were washed with phosphate buffer saline (PBS), treated with PBS containing 0.1% Triton X-100, and stained with propidium iodide (PI) (Sigma Aldrich, St. Louis, MN, USA). The DNA content was measured using FACSCalibur (Becton Dickinson, Franklin Lakes, NJ, USA). Cells were counterstained with Annexin-V-fluorescein isothiocyanate and PI using the Annexin-V-FLUOS Staining Kit (Roche, Basel, Basel-Stadt, Switzerland) for apoptosis analysis, and flow cytometric analysis was performed. The data were analyzed using the FLOWJO software (Tomy Digital Biology, Tokyo, Japan).

### 4.4. Caspase Assay

To investigate the activation capacity of Caspase-Glo 3/7 assay, Caspase-Glo 8 assay, and Caspase-Glo 9 assay (Promega, Madison, WI, USA) status of Caspase-3, -7, -8, and -9. were used. Luminescence was measured using SpectraMax iD3 (Molecular Devices, San Jose, CA, USA) 60 min after the addition of Caspase-Glo 3/7, 8, or 9 reagents to cells. The measured value was adjusted by the number of cells examined.

### 4.5. WB Analysis

WB analysis was performed as previously stated (Appendix A) [5,23,29,30,32,33]. The antibodies used are also described in Appendix A.

### 4.6. qRT-PCR

The RNeasy Mini Kit was used to isolate total RNA from NCI-H929 and OPM-2 cells (QIAGEN, Hilden, North Rhine-Westphalia, Germany). One microgram of total RNA was reverse transcribed to cDNA using a High-Capacity Reverse Transcription Kit (Applied Biosystems, Foster City, CA, USA). qRT-PCR was carried out using a QuantStudio 3 system (Applied Biosystems) to quantify the expression level of BIM. Real-time RT-PCR primer probes (BIM; Hs00708019_s1, ACTB; Hs01060665_g1) were purchased from Applied Biosystems. qRT-PCR for YPEL3, JUND, APAF1 and CDKN1B was performed using Fast SYBR Green Master Mix with a StepOne Plus instrument (Applied Biosystems). Primers were purchased from Hokkaido Systems Science Co., Ltd. (Sapporo, Japan) (Appendix A). The transcriptional levels of target genes were adjusted following the level of ACTB.

### 4.7. Microarray Analysis and GSEA

After treating NCI-H929 and OPM-2 cells with ipatasertib and/or BI-D1870 at their IC50s for 24 h, total RNA was isolated using *mir*Vana^TM^ miRNA Isolation Kit (Invitrogen, Waltham, MA, USA). GEP was analyzed with a Clariom S array (Affymetrix, Santa Clara, CA, USA), a GeneChip WT Plus Reagent Kit (Thermo Fisher Scientific, Waltham, MA, USA), and a GeneChip Scanner 7G (Affymetrix). The robust multichip array method was used to normalize the expression data. Differentially expressed genes (DEGs) were identified using one-way analysis of variance and post hoc Tukey’s honestly significant difference test after the fact. Adjusted p values for multiple comparisons were considered significant if they were less than under 0.05. Hierarchical clustering among DEGs was performed using Euclidean distance and Ward’s method. GSEA was performed using the software GSEA and the oncogenic signature gene sets from Molecular Signature Database (MSigDB) with default parameters (https://www.gsea-msigdb.org, accessed on 1 December 2021) [32,67].

### 4.8. Statistical Analysis

Statistical analyses were carried out using EZR (Saitama Medical Center, Jichi Medical University, Omiya, Saitama, Japan), a graphical user interface for R (The R Foundation for Statistical Computing, Vienna, Austria). The student’s *t*-test was used to compare continuous variables between groups, and a *p*-value less than 0.05 was considered significant.

## Figures and Tables

**Figure 1 ijms-23-02919-f001:**
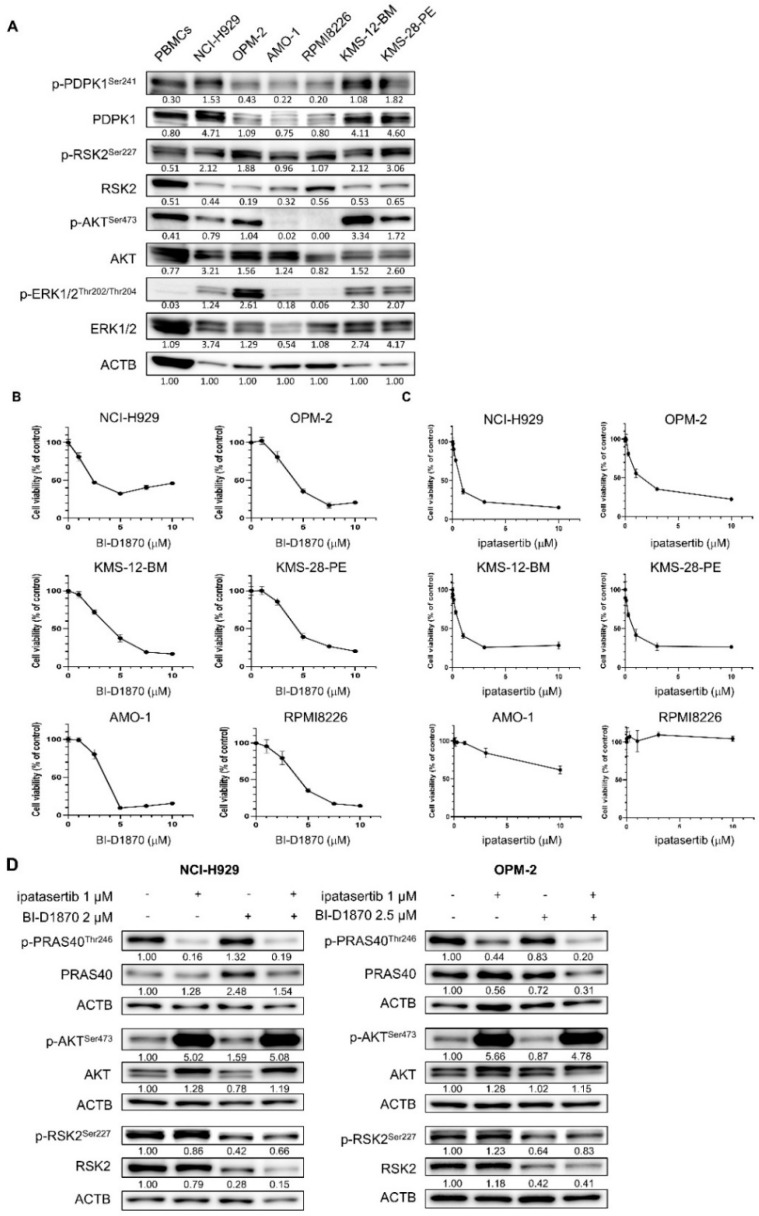
Activities and the role of cell proliferation of RSK2 and AKT in human myeloma–derived cell lines (HMCLs): (**A**) Baseline activities of RSK, AKT, and related kinases examined by Western blot (WB) in six HMCLs and peripheral blood mononuclear cells (PBMCs). The expression level of ACTB was examined as the internal control. Expression levels relative to ACTB are shown below each band measured by densitometric analysis using Image-J software. (**B**,**C**) Growth inhibitory effects of BI-D1870 (**C**) or ipatasertib (**D**) in six HMCLs. Cells were seeded at 2 × 10^5^ cells/mL and treated with various concentrations of BI-D1870 (**C**) or ipatasertib (**D**) for 48 h. The IC_50_ values of BI-D1870 for NCI-H929, OPM-2, KMS12-BM, KMS28-PE, AMO-1, and RPMI8226 cells were 4.00, 4.34, 3.88, 4.95, 3.14, and 4.09 μM, respectively, while those of ipatasertib for NCI-H929, OPM-2, KMS12-BM, and KMS28-PE cells were 0.95, 2.12, 0.81, and 0.79 μM respectively. (**D**) Effects of BI-D1870 and ipatasertib on their target molecules in NCI-H929 and OPM-2 cells. Cells were treated with either ipatasertib, BI-D1870, or their combination at the indicated concentrations for 48 h. Expression levels relative to control (untreated cells) are shown below each band measured by densitometric analysis using Image-J software. ACTB was used as an internal control.

**Figure 2 ijms-23-02919-f002:**
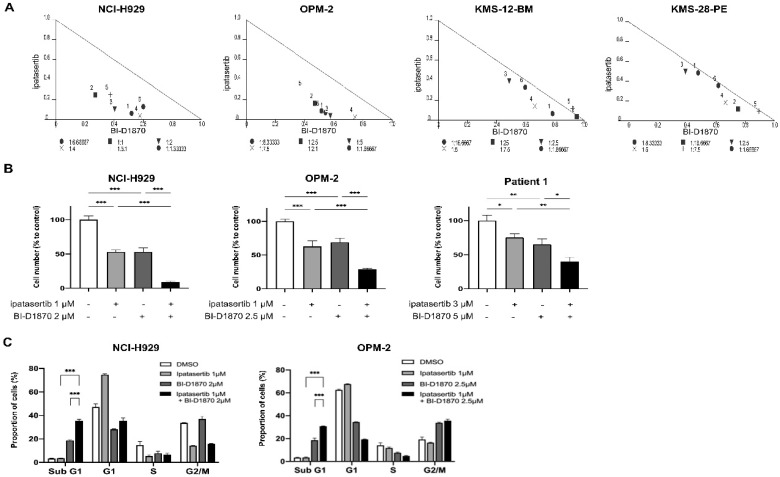
Combinatory effect of BI-D1870 and ipatasertib in HMCLs: (**A**) Isobolograms for the in vitro growth inhibitory effect by the combination of BI-D1870 and ipatasertib in NCI-H929, OPM-2, KMS-12-BM, and KMS-28-PE cells. *X*-axis and *Y*-axis indicate the fraction affected by BI-D1870 and ipatasertib, respectively. (**B**) Cells were treated with ipatasertib (1 μM) and/or BI-D1870 (2 μM) in NCI-H929, while with ipatasertib (1 μM) and/or BI-D1870 (2.5 μM) in OPM-2 for 48 h. Patient-derived myeloma cells were treated with ipatasertib (3 μM) and/or BI-D1870 (5 μM) for 48 h. (**C**–**E**) Anti-tumor effects of BI-D1870, ipatasertib, and their combination in HMCLs. Cells were seeded at 2 × 10^5^ cells/mL and treated with ipatasertib (1 μM) and/or BI-D1870 (2 μM) in NCI-H929, while with ipatasertib (1 μM) and/or BI-D1870 (2.5 μM) in OPM-2 for 48 h, and then were subjected to flow cytometric analysis. DNA content was examined by propidium iodide (PI)-staining cells for cell cycle analysis (**C**). Induction of apoptosis was examined by double staining with Annexin-V and PI. Apoptotic cells were Annexin-V (+)/PI (−) and Annexin-V (+)/PI (+) populations (**D**). The proportion of apoptotic cells were shown. Mean value with standard deviation (S.D.) of triplicate data. * *p* < 0.05, ** *p* < 0.01, *** *p* < 0.001.

**Figure 3 ijms-23-02919-f003:**
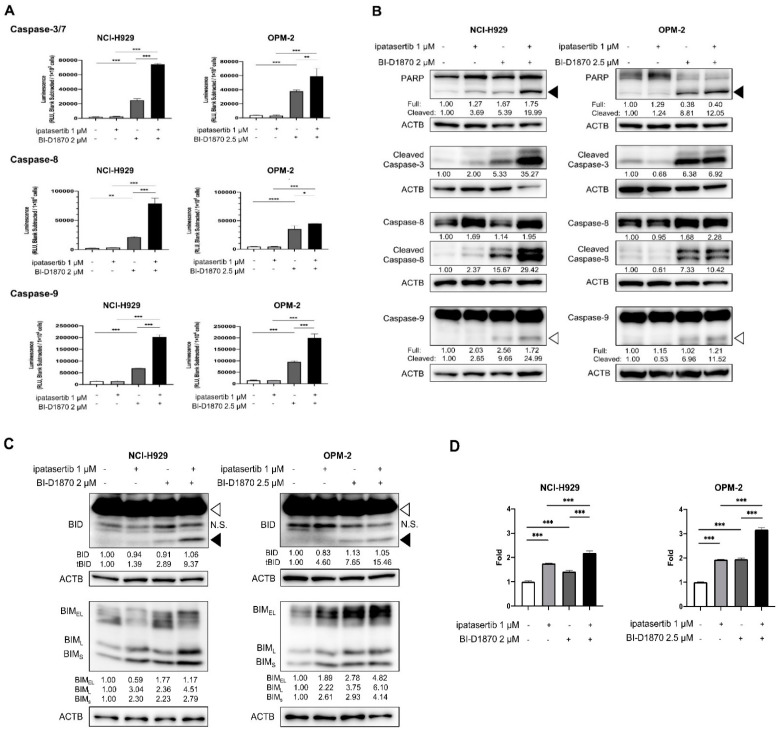
Molecular effects with apoptotic induction by BI-D1870, ipatasertib, and their combination: Cells were seeded at 2 × 10^5^ cells/mL and treated with indicated concentrations of BI-D1870, ipatasertib, or both agents for 48 h. (**A**) Activities of Caspase-3/7, Caspase-8, and Caspase-9 of untreated and treated cells measured by Caspase-Glo assay systems. (**B**) WB analyses for the detection of processing of caspases and PARP. Black arrow head and white arrow head indicate cleaved form of PARP and cleaved form of Caspase-9, respectively. (**C**) WB analysis for the detection of the activation of BID and the induction of BIM in untreated and treated HMCLs. White arrow head and black arrow head indicate full length BID and truncated BID, respectively. N.S.; no specific band. Expression levels relative to control (untreated cells) are shown below each band measured by densitometric analysis using Image-J software. ACTB was used as an internal control. (**D**) Transcriptional levels of BIM in treated cells relative to untreated cells. Mean value with S.D. of triplicate data. * *p* < 0.05, ** *p* < 0.01 *** *p* < 0.001, **** *p* < 0.0001.

**Figure 4 ijms-23-02919-f004:**
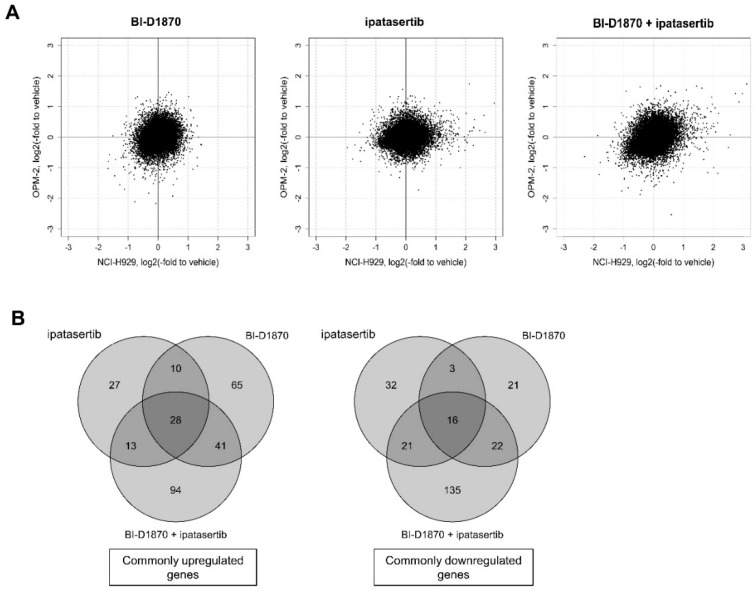
Gene expression change induced by BI-D1870, ipatasertib, or their combination in HMCLs: (**A**) Expression changes in all investigated genes relative to vehicle cells in NCI-H929 (*X*-axis) and OPM-2 cells (*Y*-axis). (**B**) Venn diagrams for numbers of commonly significantly upregulated (left) and downregulated (right) genes in NCI-H929 and OPM-2 cells. (**C**) Clustering of significantly modulated genes after treatment in NCI-H929 and OPM-2 cells.

**Figure 5 ijms-23-02919-f005:**
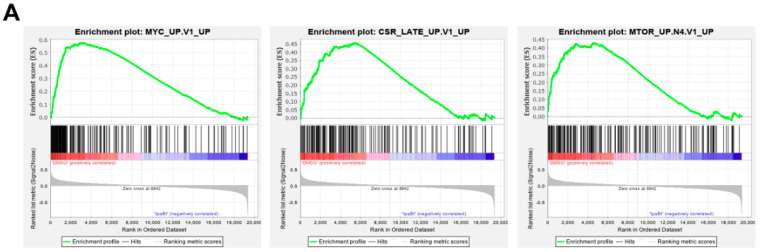
Functional assessment of gene sets significantly modulated either by BI-D1870, ipatasertib, or their combination in HMCLs: (**A**) Plots for top 3 significantly enriched gene sets, MYC_UP.V1_UP (left), CSR_LATE_UP.V1_UP (middle), and MTOR_UP.N4.V1_UP (right) (www.gsea-misgdb.org/gsea/index.jsp, accessed on 1 December 2021), consisted of genes downregulated by the combination of BI-D1870 and ipatasertib in NCI-H929 and OPM-2 cells. (**B**) List of downregulated gene sets that are more significantly enriched by the combination treatment of BI-D1870 and ipatasertib compared to the treatment by either BI-D1870 or ipatasertib alone. Significance was shown by false discovery rate (FDR)(q-value) scores. −log10(FDR-q-value) score bigger than 1.3 is significant. (**C**) Plots for top 3 significantly enriched gene sets, MTOR_UP.N4.V1_DN (left), STK33_SKM_UP (middle), and MYC_UP.V1_DN (right) (www.gsea-misgdb.org/gsea/index.jsp, accessed on 1 December 2021), consisted of genes upregulated by the combination of BI-D1870 and ipatasertib in NCI-H929 and OPM-2 cells. (**D**) List of upregulated gene sets that are more significantly enriched by the combination treatment of BI-D1870 and ipatasertib compared to the treatment by either BI-D1870 or ipatasertib alone. Significance was shown by FDR(q-value) scores. -log10(FDR-q-value) score bigger than 1.3 is significant.

**Figure 6 ijms-23-02919-f006:**
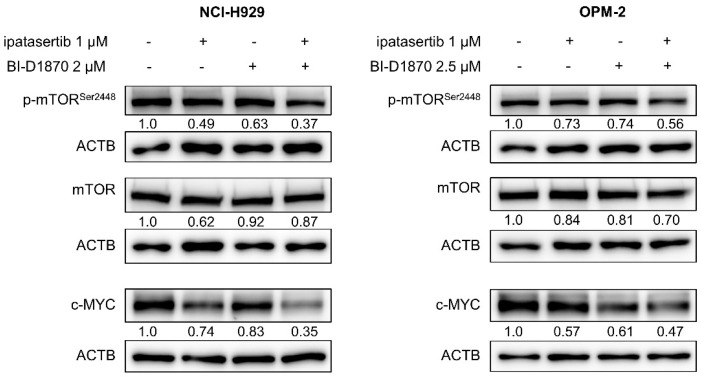
WB for MYC, mTOR, and p-mTOR: Cells were treated for 48 h at the indicated concentrations of agents. Expression levels relative to control (untreated cells) are shown below each band measured by densitometric analysis using Image-J software. ACTB was used as an internal control.

**Table 1 ijms-23-02919-t001:** List of significantly modulated genes by the treatment with the combination of BI-D1870 and ipatasertib. Gene lists of the clusters C1 and C5 in Figure 4C.

Cluster in Figure 4C	Gene Expression Change		Gene List
	Treatment		
	BI-D1870 or Ipatasertib	Combination	
C1	Significant downregulation compared to untreated cells	More significant downregulation compared to that induced by either BI-D1870 or ipatasertib	*NBPF3 PIGV STX12 OXCT2P1 BTBD8 LCE1E MGST3 UBR4 EVA1B CLDN19 PLPPR5 NBPF15 NBPF20 CBX7 LOC100131107 THBS3 GREM2 OR2T35 ACKR1 NBPF10 GBA CHIT1 MSGN1 KRTCAP3 C2orf73 BCL2L11 DNAJB2 DAW1 TGFA DLX2 PDE6D CAMP RHO ZNF445 MYL3 PRICKLE2 ZXDC C3orf36 SLC7A14 PIGZ NFKBIZ PFKFB4 POPDC2 TMEM175 BLOC1S4 USP17L19 USP17L24 MED15 USP17L25 METTL14 ADAD1 TLR10 PAPSS1 CPZ DUX4 PCDHB11 CDH12 MOCS2 MTRNR2L2 MCTP1 SLC36A2 NMUR2 TRIM7 ZNF76 NCR2 FAM83B SDHAF4 PNRC1 SMIM29 CRISP2 PLAGL1 RSPH3 ADGRF2 CAGE1 H2BC4 GABBR1 RAMP3 ZKSCAN1 DPY19L1 HERPUD2 ERVW-1 TCAF1 ZNF705D P2RX6 FAM160B2 SDCBP GPAA1 RMDN1 KCNV2 TMEM8B OR1J4 GPR21 RALGPS1 AK8 CDKN2B SYAP1 SLC16A2 TKTL1 ITIH6 KCNE5 TSPY2 WAC DPYSL4 ST8SIA6 CALHM2 AKR1C1 SMPD1 OR5L1 OR4D9 C11orf86 TTC12 C11orf40 OR51G1 OR56A5 KRTAP5-10 WNT5B C12orf54 HIGD1C SCYL2 SIRT4 HNF1A AKAP3 TAS2R42 BAZ2A APPL2 MYL6 ACAD10 CD163L1 R3HDM2 CCDC92 CRYL1 RGS6 GSKIP PRKCH RTF1 GJD2 ITGAD FOXL1 LMF1 TBX6 MTRNR2L1 RAPGEFL1 TBCD NATD1 SRCIN1 KRTAP2-3 RETREG3 ASB16-AS1 PLEKHM1 CD300LB CEP131 KRTAP9-8 PRKAR1A TBC1D3L EPG5 TNFAIP8L1 GPR42 USP29 CLEC4G BORCS8 TYROBP WDR87 CEACAM7 EYA2 CD93 ZNF334 LIPI LRRC74B UPB1 YPEL1*
C5	Significant upregulation compared to untreated cells	More significant upregulation compared to that induced by either BI-D1870 or ipatasertib	*GPR3 EIF3I UTP11 LRRC42 CFHR5 TIMM17A TMEM183A SDF4 ICMT ACOT7 MRPS15 GNL2 SF3A3 MAGOH ITGB3BP GCLM VAV3 ANP32E SLC41A1 MICOS10 SELENOI WDR43 CCT7 MRPL35 RPIA DDX18 AGPS XRCC5 ITGB1BP1 CEBPZ PNPT1 CCT4 TACR1 NIFK CHN1 HSPD1 LBH SNRNP27 CCDC138 RNASEH1 MRPS22 KCNAB1 CCDC12 NEPRO MRPL3 ANAPC13 A4GNT GK5 TMEM183B NRROS HIGD1A TWF2 ETV5 TMEM165 HADH GAR1 PLK4 NOP14 PPAT CCNA2 UFSP2 SLC39A8 CCT5 ISOC1 UNC5A OXCT1 ETF1 LARS1 GEMIN5 ZNF131 CDKAL1 MAD2L1BP NUS1 MYCT1 KCNK16 PKHD1 MMS22L SEC63 HDDC2 FBXO5 TCP1 PTP4A1 GMDS EEF1E1 NUP42 TMEM270 ASB4 DLD ST7 PRKAR1B DDX56 SBDS POM121C ASZ1 CHCHD3 SLC25A37 ZFHX4 ZNF7 C8orf33 LSM1 THAP1 UBXN8 DNAJA1 FOXB2 CENPP WDR5 MRPS2 TRUB2 MSANTD3 AWAT1 CSTF2 TCP11X2 STK26 HDAC8 MORF4L2 SERPINA7 NKRF CDY2B TXLNGY CDY1 HSPA14 PDSS1 VDAC2 LARP4B ZWINT TBATA FGFBP3 SFXN4 MRPL23 IPO7 LINC02687 P2RX3 RNF169 ACER3 DEUP1 JAM3 DEAF1 MRPL17 GALNT18 NUP160 SSRP1 ALG8 MMP3 TMX2- CTNND1 RASSF3 MSRB3 C12orf45 RIC8 FAM216A CRACR2A YARS2 IKBIP PPTC7 NDUFA9 PA2G4 PTPRQ C12orf73 MRPL57 BORA ALG5 MZT1 RUBCNL CARMIL3 AP5M1 TRMT61A ZNF219 PRMT5 PSMB5 TICRR PLA2G4F SHF TIPIN MMP25 ALDOA PSMD7 HSBP1 PDZD9 EMC8 ZNF689 CTRL PSMB6 PFAS ALKBH5 TRAF4 CISD3 C17orf80 FAM83G POLDIP2 BLMH CWC25 ETV4 ATXN7L3 CCDC182 C17orf58 CD300LF RSKR KAT2A CCDC47 GH2 EMILIN2 TIMM21 DAZAP1 CARM1 GPATCH1 WDR62 MRPS12 NR2C2AP CCNP SPINT2 CST2 WFDC9 ANKEF1 RAB5IF TGIF2-RAB5IF WDR4 SLC19A1 ATF4 BID PISD TTLL12 DRG1 SAMM50*

**Table 2 ijms-23-02919-t002:** Significantly modulated common gene sets by either BI-D1870, ipatasertib or the combination of BI-D1870 and ipatasertib in NCI-H929 and OPM-2 cells.

Downregulated by BI-D1870 Plus Ipatasertib		Upregulated by BI-D1870 Plus Ipatasertib		Downregulated by BI-D1870		Downregulated by Ipatasertib		Upregulated by Ipatasertib	
Gene Set Name	FDR *q*-Value	Gene Set Name	FDR *q*-Value	Gene Set Name	FDR *q*-Value	Gene Set Name	FDR *q*-Value	Gene Set Name	FDR *q*-Value
MYC_UP.V1_UP	0	MTOR_UP.N4.V1_DN	0	MYC_UP.V1_UP	0.0072	MYC_UP.V1_UP	0	MTOR_UP.N4.V1_DN	0.0057
CSR_LATE_UP.V1_UP	0.0012	STK33_SKM_UP	0.0058	EGFR_UP.V1_UP	0.0096	MTOR_UP.N4.V1_UP	0.0056	STK33_UP	0.0205
MTOR_UP.N4.V1_UP	0.0018	MYC_UP.V1_DN	0.0111	RPS14_DN.V1_ DN	0.0101	CSR_LATE_UP.V1_UP	0.0412	KRAS.LUNG_UP.V1_DN	0.0230
RPS14_DN.V1_DN	0.0020	STK33_UP	0.0151	CSR_EARLY_UP.V1_UP	0.0103	−	−	RPS14_DN.V1_ UP	0.0387
CSR_EARLY_UP.V1_UP	0.0039	−	−	RAF_UP.V1_DN	0.0120	−	−	−	−
VEGF_A_UP.V1_ DN	0.0045	−	−	CSR_LATE_UP.V1_UP	0.0143	−	−	−	−
ESC_V6.5_UP_ LATE.V1_DN	0.0084	−	−	ERBB2_UP.V1_ DN	0.0144	−	−	−	−
IL15_UP.V1_UP	0.0099	−	−	VEGF_A_UP.V1_DN	0.0157	−	−	−	−
ESC_J1_UP_LATE.V1_DN	0.0179	−	−	TBK1.DF_DN	0.0390	−	−	−	−
GCNP_SHH_UP_ LATE.V1_UP	0.0209	−	−	MEK_UP.V1_DN	0.0426	−	−	−	−
EGFR_UP.V1_UP	0.0234	−	−	GCNP_SHH_UP_LATE.V1_UP	0.0449	−	−	−	−
ERBB2_UP.V1_DN	0.0246	−	−	STK33_NOMO_UP	0.0456	−	−	−	−
IL2_UP.V1_UP	0.0267	−	−	−	-	−	−	−	−
MTOR_UP.V1_UP	0.0465	−	−	−	-	−	−	−	−
CAMP_UP.V1_UP	0.0468	−	−	−	-	−	−	−	−

Gene set titles are described according to www.gsea-misgdb.org/gsea/index.jsp, accessed on 1 December 2021. FDR: false discovery rate.

## Data Availability

Any data or material that support the findings in this study can be made available by the corresponding author upon request.

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
