# Peer review of "The Rationale for the Dual-Targeting Therapy for RSK2 and AKT in Multiple Myeloma"

_ijms, 2022, doi:10.3390/ijms23062919_

Round 1

Reviewer 1 Report

The present article, "The rationale for the dual targeting therapy for RSK2 and AKT 2 in multiple myeloma" is very interesting.

In this article, the authors explored therapeutic efficacy and mechanism of action of dual blockade of two major PDPK1 substrates, namely RSK2 and AKT, in Multiple myeloma. Both used drugs showed the additive to synergistic anti-tumor effect in vitro by enhancing apoptotic induction with BIM and BID activation. Moreover, the dual blockade of RSK2 and AKT exerted robust molecular effects on critical gene sets associated with myeloma pathophysiologies in vitro. The authors summarize that the dual targeting strategy for RSK2 and AKT may overcome the therapeutic difficulty due to cytogenetic/molecular heterogeneity in MM.

The article is well written. However few concerns must be addressed before publication which is vital for improving this review article. Most importantly, there is no in vivo analysis of this dual-targeting strategy.

Show IC50 value in the tabulated when cells are treated with BI-D1870, ipatasertib, and when combined.

Quantification of WB is missing in initials figures.

Does inhibitor also diminish long-term self-renewal of MM cells, like effect on clonogenicity?

Why DR4, DR5, FAS, TNFR1, FLIP, and TRAIL data is not shown? Any specific reason. Some data must be shown.

Is dual blockade of RSK2-NTKD and AKT triggers mitochondrial-dependent apoptosis to test this cytochrome c release can analyze when cells are treated with BI-D1870, ipatasertib and when combined?

Synergistic antimyeloma effects demonstrated in vitro should be confirmed in vivo. So in vivo activity of BI-D1870, ipatasertib, and when combined in a MM xenograft model, must be tested.

If possible, test growth inhibitory effects of BI-D1870, ipatasertib, and when combined in patient-derived primary myeloma cells.

Author Response

Dear Reviewer 1

We greatly appreciate your critical assessment and meaningful comments for our study. We tried our best to perform additional analyses along with the reviewers’ comments and revised the manuscript and tables, while we would like to explain for some difficulties in making reply to all queries due to both scientific reasons and the limited duration for 10 days allowed for this round of revision. Revised parts are written in red ink in the revised article.

We here would like to make replies to your comments.

Comment 1.  Show IC50 value in the tabulated when cells are treated with BI-D1870, ipatasertib, and when combined.

Reply.  We appreciate this comment and agree the reviewer’s interest. Unfortunately, we are not able to get IC50 values of BI-D1870 and ipatasertib in their combination setting with the method we utilized in this study. However, the most important point that the reviewer asked us should be to show how much degree the dose of each drug in a synergistic combination may be reduced, at a given effect level, compared with the doses of each drug alone. Therefore, instead to showing IC50 values, we would like to shown the “dose-reduction index (DRI)” of the two agents which can be analyzed with our data in a new supplementary Table S1. DRI=1 indicates no dose reduction, whereas DRI>1 and <1 indicate favorable and unfavorable dose-reduction, respectively. Our results showed the favorable DRI between ipatasertib and BI-D1870 in all four myeloma-derived cell lines. For this, we would like to add a new reference by Ting-Chao Chou (Cancer Res. 2010; 70(2) 440-446; DOI: 10.1158/0008-5472.CAN-09-1947) [66]. We hope this could be more objective and informative data for readers, and also would satisfy the reviewer’s interest.

Table S1. The dose-reduction index (DRI) with BI-D1870, ipatasertib in four RSK2-NTKD- and AKT-activated human myeloma-derived cell lines (HMCLs).

HMCL

Fraction affected

DRI

BI-D1870

ipatasertib

NCI-H929

0.787

3.70

4.01

0.911

2.47

9.48

0.967

1.67

7.65

OPM-2

0.642

1.84

13.69

0.788

2.54

3.39

0.913

1.34

1.34

KMS-12-BM

0.730

2.53

2.06

0.827

14.67

1.28

0.908

3.00

1.68

KMS-28-PE

0.359

2.11

2.05

0.578

2.563

2.01

0.801

1.63

2.81

Comment 2.   Quantification of WB is missing in initials figures.

Reply.   We thank this advice. In the revised version, we would like to add a new supplementary Figure S1 for all original Western blots with quantification data in the revised article.

Comment 3.   Does inhibitor also diminish long-term self-renewal of MM cells, like effect on clonogenicity?

Reply.   We greatly appreciate this important comment. We agree with the importance of the effect of anticancer agent on clonogenic potential of cancerous cells, and actually had some experiences of preliminary experiments for this research question; however, it was difficult for us to obtain promising finding with clonogenic assay. Indeed, we found no colony formation of several myeloma cell lines after about 10 days colony assay under the exposure to BI-D1870 at various concentrations. These results need a careful interpretation, as it was not clear whether such a finding truly demonstrated the effect on clonogenic potential by the blockade of RSK2 with even at low concentration of BI-D1870 or was just the consequence of non-specific toxic damage during long term drug exposure for about 10 days. In contrast, ipatasertib treatment had no major impacts on the number of colony formation, while the size of colony tended to be smaller compared with untreated control. With these situations, we considered that the results we obtained were too much preliminary and not promising, while we considered that the conventional clonogenic assay is mostly impossible using BI-D1870 and ipatasertib. Due to these reasons, it was also difficult for us to have the conclusive results for this comment within 10 days allow for this round of revision of our article.

Comment 4.  Why DR4, DR5, FAS, TNFR1, FLIP, and TRAIL data is not shown? Any specific reason. Some data must be shown.

Reply.  We thank this comment, and now would like to add a new supplementary Figure S3 for the WBs of DR4, DR5, FAS, TNFR1, FLIP, and TRAIL.

Comment 5.     Is dual blockade of RSK2-NTKD and AKT triggers mitochondrial-dependent apoptosis to test this cytochrome c release can analyze when cells are treated with BI-D1870, ipatasertib and when combined?

Reply.  Sorry, but it was a little bit difficult for us to understand this comment correctly. If we understood correctly, the reviewer might ask us whether cytochrome C release assay can test if apoptosis induced by BI-D1870, ipatasertib and their combination involve mitochondria-dependent intrinsic apoptosis pathway. If our interpretation is correct, the answer is yes. As shown in the text and Figures 2 and 3, our study already showed that the apoptosis induced by BI-D1870 and BI-D1870 plus ipatasertib induced apoptosis utilizing both mitochondria- and death receptor-mediated pathways, while ipatasertib alone did not cause apoptosis in myeloma cell lines. Thus, the dual blockade of RSK2 and AKT should cause cyt C relrease from mitochondria.

Comment 6.  Synergistic antimyeloma effects demonstrated in vitro should be confirmed in vivo. So in vivo activity of BI-D1870, ipatasertib, and when combined in a MM xenograft model, must be tested.

Reply.    We greatly appreciate this critically important comment, however, it was impossible for us to have this in vivo experiment due to two major reasons. One is the uncertain bioavailability of BI-D1870 for the use of animal experiment. There is no convincing data about the in vivo use of BI-D1870. In addition, although there are 55 articles that used BI-D1870, we found no experiments which investigated the effect of continuous treatment of BI-D1870 for cancer model. In addition, the other reason is that we are allowed to prepare the revised article within 10 days for this round of revision. This may also prevent us to challenge the in vivo experiments. Thus, we now would like to add/modify the sentence in the discussion section as follow: “Unfortunately, we were not able to examine the in vivo anti-myeloma effect of the combination of BI-D1870 and ipatasertib, and this was the limitation of the current study. To implement this strategy in a clinical environment, the clinical development of bioavailable dual inhibitors for RSK2 and AKT is expected for MM [63].”

Comment 7.    If possible, test growth inhibitory effects of BI-D1870, ipatasertib, and when combined in patient-derived primary myeloma cells.

Reply.  We again appreciate this important comment. As mentioned above, we are allowed to prepare the revised article within 10 days for this round of revision. In addition, we have only 5 working days during the given 10 days. Unfortunately, we had no chance for obtaining a fresh sample of patient-derived myeloma cells during this period (from 14th to 18th, February), and were not able to add the suggested experiment.

We again thank your thoughtful and excellent suggestions and comments. We believe that the revised article becomes much better than the original version by the reviewers’ suggestions. Your consideration will be much appreciated, and I look forward to hearing from you.

Sincerely yours,

Junya Kuroda M.D., Ph.D.

 Professor and Chair

Division of Hematology and Oncology,

Kyoto Prefectural University of Medicine

465, Kajii-cho, Kamigyo-ku, Kyoto, Japan

E-email: junkuro@koto.kpu-m.ac.jp

Reviewer 2 Report

Reiko Isa et al. uncovered the rationale of RSK2 and AKT targeting in multiple myeloma.

Points to be addressed

  1.  for all Western blot figures, densitometry readings/intensity ratio of each band should be included; the whole Western blot showing all bands and molecular weight markers should be included in the Supplementary Materials
  2. The authors state "To combat MM which has remarkable cytogenetic/molecular heterogeneity among patients and intraclonal diversity even in a single patient, a series of therapeutic strategies have been developed over the past two decades. Thanks to major players including proteasome inhibitors, immune therapeutics immunomodulatory drugs (IMiDs), and monoclonal antibodies against CD38. It would be good to boost the state of the art and introduce much better a detailed description of the methodology used in light of the rationale: methodologies, objectives and results that the manuscript aimed to achieve and its actual success in achieving them, with a focus dor the advancement of the available knowledge.

    3. A short overview (workflow or graphical abstract) regarding the modalities of integration of the different approaches used

    4. I miss possible application potentialities and/or technological and clinical real-life impact

    5. The authors mentioned the cell adhesion role, I quote "cell-extrinsic tumor microenvironment supports the survival and proliferation of myeloma cells via adhesion molecules, soluble factors, or extracellular vesicles, and also protects myeloma cells from cytotoxic insults"and reference correctly reff 6,16,17. Recent data pointed out that the expression levels of junctional adhesion molecules (JAMs) and JAM-A in particular, by malignant plasma cells can predict disease outcomes. Moreover, elevated membrane expression of JAM-A also on bone marrow endothelial cells predicted poor outcomes. Those data can nicely match with the authors' findings regarding AKT targeting and should be included (please refer to PMID: 32354870).

Author Response

Dear Reviewer 2

We greatly appreciate your critical assessment and meaningful comments for our study. We tried our best to perform additional analyses along with the reviewers’ comments and revised the manuscript and tables, while we would like to explain for some difficulties in making reply to all queries due to both scientific reasons and the limited duration for 10 days allowed for this round of revision. Revised parts are written in red ink in the revised article.

We here would like to make replies to your comments.

Comment 1.  For all Western blot figures, densitometry readings/intensity ratio of each band should be included; the whole Western blot showing all bands and molecular weight markers should be included in the Supplementary Materials.

Reply.  We thank kind advice, and would like to add densitometric data for all WBs, and also add a new supplementary Figure S1 for all original Western blots with quantification data in the revised article.  Unfortunately, it was difficult for us to show molecular wight markers in all figures for WB due to the system we used. Please find the photo attached below (please see the attached WORD file) for one example of our WB assays in this study. With our system for the detection of WB bands, we evaluated WB bands and also confirmed the band size using molecular weight markers using 3 different modes (Mode 1-3) for one membrane. With this system, we especially confirmed the band size using Mode 1 and Mode 2 (red squares), and, then, used the figure shown by Mode 3 as the final results of experiments in our manuscript. With the Mode 3, the molecular weight markers become very faint, mostly invisible white bands (should be present in blue squares). This is the reason for the difficulty in showing molecular weight markers in WB figures. We hope that the reviewer kindly accepts this explanation, and confirms the validity of our assays.

Comments 2-4.

Comment 2.   It would be good to boost the state of the art and introduce much better a detailed description of the methodology used in light of the rationale: methodologies, objectives and results that the manuscript aimed to achieve and its actual success in achieving them, with a focus or the advancement of the available knowledge.

Comment 3.  A short overview (workflow or graphical abstract) regarding the modalities of integration of the different approaches used

Comment 4.    I miss possible application potentialities and/or technological and clinical real-life impact.

Reply.  We greatly appreciate these kind advices, and would like to answer to comments 2-4 together. If we understood these comments correctly, the reviewer encourages us to add more discussion about the clinical role/position of the dual targeting of RSK2 and AKT for overcoming the resistance to currently available therapeutics with functional relevance, and also the future possible clinical application. To reply this, we would like to add the following considerations with several new references (#58-62) in the Discussion section: “Despite the remarkable improvement of prognosis of patients with MM by the advent of combinatory therapies of PIs, IMiDs, and monoclonal antibodies, MM still remains to be mostly incurable. Therefore, the therapeutic strategy which can overcome the resistance to the combination of the three class agents is an unmet medical need in MM. Reportedly, the incidence of mutations within the RAS/ERK pathways is significantly increased in MM refractory to PIs and IMiDs [58]. Mechanistically, oncogenic activation of RAS/RAF/ERK signaling is one of the mechanisms that enhance proteasome capacity, while treatment with PI causes the phosphorylation of AKT [59, 60], and both mechanisms contribute to the resistance to proteasome inhibition in myeloma cells. AKT-mediated sustained eIF4E expression and C/EBP translation have been proposed as the resistant mechanisms for IMiDs [61]. Moreover, the resistant mechanisms for current standard therapeutics for MM include aberrant MYC overexpression [50], and the increased rate of protein translation [62]. Considering these, one of the clinical positions of the dual targeting therapy for RSK2 and AKT may be theoretically the salvage setting after treatment failure of PIs, and/or IMiDs with or without monoclonal antibodies for CD38 or SLAMF-7.”

We also would like to version up the graphical abstract (below) which include the information about the role of the direct dual blockade approach for RSK2 and AKT and the targets of different therapeutic approaches.

Graphical abstract

 (please see a new graphical abstract on the WORD file attached)             

Comment 5.  Recent data pointed out that the expression levels of junctional adhesion molecules (JAMs) and JAM-A in particular, by malignant plasma cells can predict disease outcomes. Moreover, elevated membrane expression of JAM-A also on bone marrow endothelial cells predicted poor outcomes. Those data can nicely match with the authors' findings regarding AKT targeting and should be included (please refer to PMID: 32354870).

Reply.   We thank this kind advice, and now would like to add the suggested article as a new reference #18.

We again thank your thoughtful and excellent suggestions and comments. We believe that the revised article becomes much better than the original version by the reviewers’ suggestions. Your consideration will be much appreciated, and I look forward to hearing from you.

Sincerely yours,

Junya Kuroda M.D., Ph.D.

Professor and Chair

Division of Hematology and Oncology,

Kyoto Prefectural University of Medicine

465, Kajii-cho, Kamigyo-ku, Kyoto, Japan

E-email: junkuro@koto.kpu-m.ac.jp

Round 2

Reviewer 1 Report

The authors addressed all my comments, and I have no more suggestions. 
Although I feel to improve the novelty of this work, both in vivo and some experiments on patient-derived primary myeloma cells must be performed. 

Author Response

Dear Reviewer 1

We greatly appreciate your critical assessment and meaningful comments for our study. We tried our best to perform additional analyses along with your comments and revised the manuscript and tables, while we would like to explain for some difficulties in making reply to all queries due to both scientific reasons and the limited duration for 10 days allowed for this round of revision. Revised parts are written in red ink in the revised article.

We here would like to make replies to your comments.

Comment 1. The authors addressed all my comments, and I have no more suggestions. Although I feel to improve the novelty of this work, in vivo and experiments on patient-derived primary myeloma cells must be performed. 

Reply. We appreciate the important comment. We really agree with your opinion about the importance of the work with patient-derived sample and in vivo experiment. We tried our best to make prompt reply to the reviewer 1 within 10 days, including 6 working days, allowed for the preparation of the revised article.                      

As for the experiments with patient-derived sample, we were able to have primary myeloma cells from one patient with relapsed/refractory myeloma who acquired the resistance to multiple types of proteasome inhibitors, immunomodulatory drugs, and monoclonal antibodies. We would like to add (below) the result of in vitro effects of BI-D1870, ipatasertib, and their favorable combination in the revised Figure 2B in this revised article (please see WORD file for figure). It was our best to have one patient who needed bone marrow analysis in the daily clinical practice and agreed with the use of his myeloma cells for our research in 6 working days allowed for the preparation of this revision.

For the animal model study, however, I again would like to explain the reason for impossibility of in vivo animal model, as was also the case with the first-round revision.

What we found with the role of RSK2 is its functional importance of the activity of N-terminal kinase domain (NTKD) in myeloma cell survival and proliferation, and BI-D1870 is the only commercially available convincing inhibitor for RSK2-NTKD. It is really unfortunate for us that in vivo bioavailability of BI-D1870 is uncertain and not validated, therefore, we have no chance for in vivo experiments with BI-D1870, while some of other inhibitors targeting C-terminal kinase domain of RSK2 may be available in vivo.

Because we really agree with your opinion in the first-round revision, we added some discussion in the Discussion part as follow: “Unfortunately, we were not able to examine the in vivo anti-myeloma effect of the combination of BI-D1870 and ipatasertib, and this was the limitation of the current study. To implement this strategy in a clinical environment, the clinical development of bioavailable dual inhibitors for RSK2 and AKT is expected for MM [63].” However, this was not fully sufficient.

In this second-round revision, we would like to further add the explanation about the uncertain bioavailability of BI-D170 as follow: “Unfortunately, we were not able to examine the in vivo anti-myeloma effect of the combination of BI-D1870 and ipatasertib, due to the uncertain in vivo performance of BI-D1870, and also the lack of promising in vivo bioavailable other inhibitor specific for RSK2-NTKD. This was the limitation of the current study, which urges the generation of in vivo bioavailable inhibitor for RSK2-NTKD. To implement this strategy in a clinical environment, the clinical development of bioavailable dual inhibitors for RSK2 and AKT is expected for MM [63].”

As referred with the reference article No. 63, we actually started the work with TAS-0612 which is bioavailable dual inhibitor for RSK2 and AKT, based on the findings obtained in this study. Along with your advice, we would like to analyze the effect of the new compound on patient-derived myeloma cells from large number of patients, and also would like to perform in vivo experiments for the future work. However, it would take at least one year for the work with TAS-0612, including studies with in vivo animal model, and, therefore, we would like to publish current work as the current form to demonstrate the molecular rationale of the dual targeting therapy for RSK2-NTKD and AKT.

I hope that the reviewer 1 would kindly understand the current situation of commercially available RSK2-NTKD inhibitors and understand that the present work is the one which really shows the molecular rationale for the development of dual targeting therapy for RSK2 and AKT in myeloma. Based on this fundamental work, we would like to move to the next challenge for the clinical development of the dual targeting therapy.

We again thank your thoughtful and excellent suggestions and comments. We believe that the revised article becomes much better than the original version by the reviewers’ suggestions. Your consideration will be much appreciated, and I look forward to hearing from you.

Sincerely yours,

Junya Kuroda M.D., Ph.D.

Professor and Chair

Division of Hematology and Oncology,

Kyoto Prefectural University of Medicine

465, Kajii-cho, Kamigyo-ku, Kyoto, Japan

E-email: junkuro@koto.kpu-m.ac.jp

Reviewer 2 Report

The authors have clarified several of the questions I raised in my previous review. Most of the major problems have been addressed by this revision.

Author Response

Dear Reviewer 2

We greatly appreciate your critical assessment and meaningful comments for our study. We here would like to make reply to your comments.

Comment 1.  The authors have clarified several of the questions I raised in my previous review. Most of the major problems have been addressed by this revision.

Reply  We again thank your thoughtful and excellent suggestions and comments at the first-round revision. We believe that the revised article becomes much better than the original version by your suggestions.

Your consideration will be much appreciated, and I look forward to hearing from you.

Sincerely yours,

Junya Kuroda M.D., Ph.D.

Professor and Chair

Division of Hematology and Oncology,

Kyoto Prefectural University of Medicine

465, Kajii-cho, Kamigyo-ku, Kyoto, Japan

E-email: junkuro@koto.kpu-m.ac.jp

Round 3

Reviewer 1 Report

I have no more comments.